# Experimental Research on NO_2_ Viscosity and Absorption for (1-Ethyl-3-methylimidazolium Trifluoroacetate + Triethanolamine) Binary Mixtures

**DOI:** 10.3390/molecules26226953

**Published:** 2021-11-18

**Authors:** Baoyou Liu, Xinyu Wang, Jie Tian, Peiwen Zhang, Huilong Yang, Nanxi Jin

**Affiliations:** 1Pollution Prevention Biotechnology Laboratory of Hebei Province, College of Environment Science and Engineering, Hebei University of Science and Technology, Shijiazhuang 050018, China; a1083395088@163.com (X.W.); commonhao97@gmail.com (J.T.); a13315589338@163.com (N.J.); 2General Affairs Office of Hebei University of Science and Technology, Hebei University of Science and Technology, Shijiazhuang 050018, China; yanghl353@163.com

**Keywords:** ionic liquid, 1-ethyl-3-methylimidazolium trifluoroacetate, triethanolamine, viscosity, NO_2_ absorption

## Abstract

The viscosity (9.34–405.92 mPa·s) and absorption capacity (0.4394–1.0562 g·g^−1^) of (1-ethyl-3-methylidazolium trifluoroacetate + triethanolamine) binary blends atmospheric pressure in the temperature range of 303.15–343.15 K and at different mole fractions of [EMIM] [TFA] have been carried out. The molar fraction of [EMIM] [TFA] dependence of the viscosity and absorption capacity was demonstrated. The addition of a small amount of [EMIM] [TFA] into TEA led to rapidly decreased rates of binary blends’ viscosity and absorption capacity. However, the viscosity and absorption of binary blends did not decrease significantly when [EMIM] [TFA] was increased to a specific value. Compared with the molar fraction of the solution, the temperature had no obvious effect on viscosity and absorption capacity. By modeling and optimizing the ratio of viscosity and absorption capacity of ([EMIM] [TFA] + TEA), it is proven that when the mole fraction of [EMIM] [TFA] is 0.58, ([EMIM] [TFA] + TEA) has the best viscosity and absorption capacity at the same time. In addition, at 303.15 K, ([EMIM] [TFA] + TEA) was absorbed and desorbed six times, the absorption slightly decreased, and the desorption increased.

## 1. Introduction

Nitrogen oxides (NO_X_) are the prominent pollutants in the atmosphere. They are mainly produced by the high-temperature combustion of industrial nitrogenous fuel in which NO_X_ emission accounts for 50% of total emission when burning petroleum and over 60% (about 7.7 million tons) when burning coal in power plant boilers [1,2,3]. NO_X_ are the main precursors of acid rain and the direct sources of indoor nitrous acid (HONO) and photochemical air pollution. They greatly increase the risk of lung damage or even lung cancer if exposed to these substances for a long time [4,5,6]. At present, SCR (selective catalytic reduction), SNCR (selective non-catalytic reduction), and NSCR (non-selective catalytic reduction) are the principal force of denitrification techniques. However, these technologies are often accompanied by high technology costs and complex tail gas pollution. In contrast, inexpensive solvents are becoming more attractive. The absorbents (water, acid solution, alkali solution) which target nitrogen dioxide (NO_2_) account for the majority of these solvents [7,8]. As a mature and efficient separation system for industrial desulfurization and denitrification, alkanolamines (such as monoethanolamine, diethanolamine, methyldiethanolamine, and triethanolamine) can rapidly transfer acid gas from gas phase to liquid phase and have significant advantages of high absorption and low operation costs [9,10,11]. The mixture of alkanolamines and water is a common form of industrial solvents. The cavities of the water provide larger activity spaces for the molecules of alkanolamines and the two-phase hydrogen bonds significantly prevent the self-association between the molecules [12]. This behavior effectively reduces the viscosities of pure alkanolamines, strengthens the mass transfer, and increases the absorption capacity of gases. However, the degradation and volatilization of alkanolamine solutions—as well as the scaling and corrosion of equipment carried by the water of the compound system—are still unavoidable in the industrial process [13]. The crucial problem is that NO_2_ reacts with alkanolamine solutions based on water easily and rapidly which makes the solutions become viscous and foam, at the same time causing the system to generate higher regeneration energy, which challenges the practicability and circulation of the absorption solutions in large absorption equipment [14,15]. Some researchers have carried out a lot of work on the absorption of acid gas by nonaqueous amine solutions [16,17,18]. For example, Kim et al. [17] used alcohol solvent as a substitute for water. Compared with water or short-chain alcohol, the use of (long-chain alcohol + alkanolamines) solution reduces the wastage of high volumes of solvents, but the insufficient absorption capacity in the process still needs to be further studied and solved in the future. Therefore, finding a safe and stable solvent is necessary to enhance the process efficiency further and reduce the environmental impact caused by the volatilization of absorbents.

In recent years, as one of the hotspots in the field of green chemistry, ionic liquids (ILs) have been proposed as new candidate materials for selective capture of CO_2_, SO_2_, and H_2_S due to their unique physical and chemical properties, such as adjustable structures, low vapor pressure, high thermal stability and chemical stability, high solubility for organic and inorganic compounds, and wide liquid phase range [18,19,20]. ILs have also been studied to absorb NO_2_ in recent reports. Kunov Kruse et al. [21] first analyzed the feature spectra of several imidazolyl ILs for capturing NO_2_. These ILs can easily overcome the Lewis base catalytic reaction between NO_2_ and ILs in an anhydrous state, which prevents the solvent from anion loss and dramatic viscosity changes at high temperatures. Yuan et al. [22] found that NO_2_ exists as a dimer among the molecules of [BMIM] [OTF]. Processes similar to molecular-imprinting of NO_2_ during absorption–desorption illustrate the adsorption of NO_2_ in ILs is controllable and reversible, and the absorption quantity can reach 0.0322 g·g^−1^ at the normal state. Duan et al. [20] tested the solubility of pure NO_2_ in the special amine-functionalized ILs. The best among these ionic liquids was CPL:TBAB (2:1) whose absorption quantity was 0.355 g·g^−1^, about 10 times of the traditional imidazolyl ILs. It can be concluded that amine-functionalized ILs have similar characteristics to those of organic amine solutions. However, amine-functionalized ILs have fewer basic amine groups than alkanolamines, and their anions easily interact with the tail of cation (—NH_2_) through hydrogen bonds, which makes the viscosities of pure amine-functionalized ILs relatively high, further leading to lower gas capture capacity in the industry [23,24]. In this case, ionic liquids used alone are not enough to separate NO_2_ maturely and economically.

In recent years, researchers have tried to use binary mixture (ionic liquid + alkanolamine) to absorb acid gas [25,26,27,28,29,30]. It has been found that the mixing of binary systems significantly increases the absorption of acid gas, reduces the possible corrosion to equipment, and reduces the viscosity of the fluid, which is very important. The method can reduce pumping energy consumption and the loss in the flow process and improve equipment durability. However, the absorption of NO_2_ by binary mixtures (ILs + alkanolamines) has yet to be studied. According to Wolf et al. [31], based on imidazole ionic liquids in 2012, and Gold et al. [32] on the good trapping effect of triethanolamine on NO2 in aqueous solution. Therefore, we prepared neutral ionic liquid 1-ethyl-3-methylimidazole trifluoroacetate ([EMIM] [TFA]) and mixed it with triethanolamine (TEA) in different proportions, and further studied the relationship between NO_2_ absorption and viscosity in the binary system. The experimental temperatures of ([EMIM] [TFA] + TEA) are 303.15 K, 313.15 K, 323.15 k, 333.15 k, and 343.15 k, respectively.

## 2. Results and Discussion

### 2.1. Viscosity and Model

The viscosity of TEA and [EMIM] [TFA] in 298.15 K–333.15 K and their comparison with the reported literature values are shown in Table 1. Figure 1 shows that the viscosity measurement values of TEA are basically consistent with the values of Lopez et al. [33] and Li et al. [34,35], with an average error of 0.7985%. The viscosity values of [EMIM] [TFA] are slightly lower than those of Karim et al. [36] and Hector et al. [37], with an average error of −3.512%. This phenomenon may be due to the extreme sensitivity of [EMIM] [TFA] to chloride ions and trace moisture (>104 ppm) carried in the synthesis process [38].

The viscosity values of ([EMIM] [TFA] + TEA) binary mixtures at different temperatures and molar fractions are shown in Table 2 and Figure 2. It can be seen that the viscosity of binary mixture decreases with the increase in temperature, and the viscosity of the mixture is always between the two-phase pure liquid, that is, η_TEA_ < η_([EMIM] [TFA] + TEA)_ < η_[EMIM] [TFA]_. The viscosity of [EMIM] [TFA] decreases only 2.84 times in the temperature T range 303.15 K–353.15 K, while the viscosity of TEA decreases 11.18 times. TEA is more susceptible to temperature variation than [EMIM] [TFA], which can be attributed to the fact that a large number of intermolecular forces in TEA decrease with the increase in temperature [34]. Pseudo-quantitative changes of viscosity η with temperature and composition for the ([EMIM] [TFA] + TEA) binary systems are shown in Figure 3. The viscosity of binary mixtures ([EMIM] [TFA] + TEA) (9.34–405.92 mpa·s) is more susceptible to the influence of the molar fraction rather than temperature, which is consistent with the result of Haghtalab et al. [41]. Viscosity is always negatively correlated with the molar fraction χ_1_. The viscosity of the binary system decreases rapidly in the enrichment regions of TEA (0 < χ_1_ < 0.5), and the viscosity changes slowly in the enrichment regions of [EMIM] [TFA] (0.5 < χ_1_ < 1). The relationship between the viscosity of the binary mixture ([EMIM] [TFA] + TEA) and the molar fraction of [EMIM] [TFA] is well fitted by the sixth order polynomial Equation (1), and the fitting parameters are shown in Table 3.
H = A_0_ + A_1_x + A_2_x^2^ + A_3_x^3^ + A_4_x^4^ + A_5_x^5^ + A_6_x^6^(1)

The viscosity deviation values were calculated from the data of viscosity by using the following Equation (2).
(2)$$\Delta \eta \;=\;\eta \;-\;\chi_{1}\eta_{1}\;-\;\chi_{2}\eta_{2}$$
where *η*, *η*_1_ and *η*_2_ stand for the viscosity of ([EMIM] [TFA] + TEA), pure [EMIM] [TFA] and pure TEA, respectively. *χ*_1_ and *χ*_2_ stand for the molar fraction of [EMIM] [TFA] and TEA. The calculations are shown in Table 4 and the relationship between the molar fraction of [EMIM] [TFA] and viscosity deviations ∆*η* is shown in Figure 4.

Similar to the behavior of most non-aqueous binary mixtures [41,42,43], the viscosity deviation of ([EMIM] [TFA] + TEA) binary mixtures presents an entirely negative deviation which is different from the ideal situation in the whole range of χ_1_ in Figure 4. This behavior can be explained by weakening strong and specific interactions in the blends [44]. At the same time, it was observed that with the increase in [EMIM] [TFA] molar fraction, the viscosity deviation first decreased and then increased. When χ_1_ is about 0.4, the absolute viscosity deviation is max, and the maximum value is |∆*η*_min_|. Meanwhile, |∆*η*_min_| decreases from 175.48 mPa·s to 12.19 mPa·s with temperature (303.15 K–43.15 K). Furthermore, the viscosity deviations have also been fitted by a Redlich–Kister [45] type Equation (3), a fourth-order polynomial was found to be optimum for viscosity deviations of binary mixtures ([EMIM] [TFA] + TEA). The standard deviation was calculated by using the following Equation (4). The fitted parameters and standard deviation σ were listed in Table 5.
(3)$$\Delta Q\;=\;X_{1}(1\;-\;X_{1}){\sum_{P\;=\;0}^{M}B_{P}(1\;-\;2X_{1})}^{P}$$
(4)$$\sigma \;=\;{\left[\sum_{1}^{n_{dat}}{\left(Z_{\exp}\;-\;Z_{cat}\right)}^{2}/n_{dat}\right]}^{1/2}$$
where ∆*Q* represents ∆*η* is the excess property, *χ*_1_ is the molar fraction of [EMIM] [TFA], *B_p_* is the fitting parameters, M is the degree of the polynomial expansion, which was optimized using the Marquardt algorithm.

where *n_dat_* is the number of experimental data points, *Z*_exp_ and *Z_cat_* are the experimental value, calculated by Equation (3), respectively.

### 2.2. Absorption Capacity

Table 6 shows the absorption of NO_2_ by different molar fractions ([EMIM] [TFA] + TEA) from 303.15 K to 343.15 K. A second-order polynomial Equation (5) was found to satisfactorily correlate the change of absorption with temperature. By fitting these data, it is found that the second-order polynomial Equation (5) has a good correlation with the change of absorption with temperature. The parameters of the equation are shown in Table 7. ([EMIM] [TFA] + TEA) and NO_2_ absorption is negatively correlated with temperature, as shown in Figure 5. Even though the viscosity of blends is low at high temperatures, the low physical solubility of ([EMIM] [TFA] + TEA) seems to dominate. The change of absorption with different molar fractions of [EMIM] [TFA] at specific temperatures and at one atmosphere is shown in Figure 6 and Figure 7. With the increase in the TEA component, the absorption of NO_2_ increased. It can be considered that the mole fraction of TEA is significant for the interaction between the binary mixture of ([EMIM] [TFA] + TEA) and NO_2_. The relationship between the mole fraction of [EMIM] [TFA] and the absorption is well fitted by the fourth-order polynomial Equation (6), and the fitting parameters are shown in Table 8.
A = C_0_ + C_1_t + C_2_t^2^(5)
A = D_0_ + D_1_x + D_2_x^2^ + D_3_x^3^ + D_4_x^4^(6)

With the increase in the molar fraction of TEA in the binary mixture ([EMIM] [TFA] + TEA), the absorption of NO_2_ increased. However, at the same time, the viscosity of the binary system increases rapidly, which is easy to make the absorption equipment stagnate and lead to wall adhesion. For example, at 313.15 K, the viscosity of TEA is 207.24 mPa s, which is about 12 times the viscosity of pure ionic liquid under the same conditions. Therefore, the problem of excessive viscosity should be avoided while ensuring excellent absorption. By fitting the relationship between the [EMIM] [TFA] molar fraction, absorption, and viscosity of ([EMIM] [TFA] + TEA) solution, a mathematical model was established with the minimum ratio of viscosity to the absorption as the optimization objective. The solution was optimized by Origin, and the optimal ([EMIM] [TFA] + TEA) molar fraction was carried out under the higher absorption and lower viscosity. The optimization diagram is shown in Figure 8.

Taking the result at 303.15 K as an example, the results show that the function has a minimum value of 44.80 when the molar fraction of [EMIM] [TFA] is 0.58, the viscosity is 35.51 mPa·s, and the adsorption capacity is 0.7927 g·g^−1^, at this time ([EMIM] [TFA] + TEA) has a lower viscosity and a higher adsorption capacity. With the increase in molar fraction (*χ*_1_), the ratio of viscosity to absorption almost reached equilibrium. In order to verify the absorption effect of the experimental data ([EMIM] [TFA] + TEA) on NO_2_, we compared it with the ratio of viscosity value and absorption quality of NO_2_ absorbents prepared in the existing literature (at the same temperature and pressure). MDEA (101.00 mPa·s/0.021 g·g^−1^ = 4809.00) [10]; CPL-TBAB (6502.38 mPa·s/0.355 g·g^−1^ = 18,316.56) [20,21]; [BMIM] [OTF] (132.70 mPa·s/0.032 g·g^−1^ = 4146.88) [22,46]. The absorption capacity of NO_2_ by the mixture of ionic liquid created in this experiment is significantly better than those of other absorbents, and the negative effect of water was eliminated.

### 2.3. Theoretical Analysis

Spectroscopic studies show that there is a reversible equilibrium between NO_2_ and N_2_O_4_, (the partial pressures of N_2_O_4_ at different temperatures are shown in the Supplementary Data [47,48]), and NO^+^ and NO_3_^−^ are easily produced by autoionization in liquids or organic phases [49]. The molecular adduct [(Don)_n_·NO^+^]NO_3_^−^ will be formed reversibly When N_2_O_4_ is absorbed by the organic solvent (Don) [49]. If *n* > 2, the molecular adduct is more stable. Based on the experimental phenomena and theoretical analysis, we thought that the reaction of NO_2_ in pure TEA is similar to that introduced by Addison, and the reaction product is a stable ionic substance [{(HOC_2_H4)_3_N}_2_·NO^+^]NO_3_^−^, as shown in Figure 9. We found that the system’s viscosity tends to decrease when adding ionic liquid [EMIM] [TFA]. The viscosity of the binary mixture drops sharply when χ_1_ = 0–0.5, which is similar to the mixing of H_2_O and TEA. It can be attributed to the fact that [EMIM] [TFA] provides a large number of cavities for TEA, which hinders the self-association of TEA molecules. However, [EMIM] [TFA] has little effect on the binary system except for the absorption decrease in the range of χ_1_ = 0–1. Compared with the characteristic spectrum of pure ionic liquid, there are no new peaks and relative peak displacements in the characteristic spectrum of the regenerated [EMIM] [TFA] (Figure 10). Therefore, it is considered that [EMIM] [TFA] only has physical absorption in ([EMIM] [TFA] + TEA) and does not change the properties of the system.

### 2.4. Reusability of ([EMIM] [TFA] + TEA)

The recovery rate and reuse effect of ([EMIM] [TFA] + TEA) can be determined by absorption and desorption experiments. Therefore, the purpose of the technical efficiency and economic feasibility can be realized. In this work, the reusability of ([EMIM] [TFA] + TEA) was studied. Usually, saturated absorption amount was observed in 60 min at 303.15 K and 1.01 × 10^5^ Pa, with a stream of 30 mL·min^−1^ pure NO_2_, and the captured NO_2_ can be removed in 40 min at 70 ℃ under vacuum (4.24 kPa). When the molar fraction of [EMIM] [TFA] was 0.58, a recycle of six times NO_2_ absorption and desorption of ([EMIM] [TFA] + TEA) at 303.15 K was observed (Figure 11). The quantitatively recovered ([EMIM] [TFA] + TEA) was directly used in the following absorption process without further treatment. The results showed that the absorption was highly reversible, and the absorption amount remains basically unchanged after six consecutive absorption/desorption cycles. The slight decrease in absorption may be due to the loss of TEA and the increase in ILs mole fraction., but the advantage of this situation was that the desorption quantities of NO_2_ were increased.

## 3. Materials and Methods

### 3.1. Materials

1-Methylimidazole (≥98%, Shijiazhuang Sydano Fine Chemical Co. Ltd, Shijiazhuang, China); silver trifluoroacetate (≥99%, Tianjin Hines Biochemical Technology Co. LTD, Tianjin, China); bromoethane (≥99%, Tianjin Damao chemical reagent plant, Tianjin, China); acetonitrile (≥99%, Tianjin Bodi Chemical Co. LTD, Tianjin, China); triethanolamine (≥99%, Tianjin Hedong District Hongyan reagent plant, Tianjin, China); NO_2_ (≥99.5%); NaOH (≥96%, Xinkou Industrial Park, Xiqing Development Zone, Tianjin, China). All drugs are commercially available and have no further purification (See Table 9 for details of samples). [EMIM] [TFA] was prepared similarly to the literature method [50,51]. 1-methylimidazole (30 mL) and ethyl bromide (90 mL) were fully mixed in a 250 mL round bottom flask and heated under reflux for 8 h to obtain an oily liquid. Then, the liquid was cooled to room temperature and a white solid was obtained. Then, the white solid was dissolved using acetonitrile and filtered, the ethyl acetate was added to the filtrate to obtain a white solid. The above operations were repeated, and the recrystallized product was dried under vacuum for 36 h to obtain 1-ethyl-3-methylimidazolium bromide (EMIMBr). Then, 19.1 g of EMIMBr and 22.4 g of silver trifluoroacetate were weighed out and added to 200 mL of acetonitrile. Finally, the silver bromide precipitate was filtered and the acetonitrile was removed by rotary evaporation. The final product [EMIM] [TFA] was obtained after vacuum drying at 60 ℃. The reaction process is shown in Figure 12. The structure spectrum of [EMIM] [TFA] is as follows. IR film, cm^−^^1^: C=N (3374.54 cm^−1^); -CH_2_-(2941.08 cm^−1^); C=O (1683.35 cm^−1^); C=C(1572.82 cm^−1^); -CH_2_-(1456.16 cm^−1^); C-F(1065.90 cm^−1^). ^1^H NMR(500 MHz, DMSO-d6, δ, ppm): 1.364 ppm (t, 3H, -(CH_2_)-CH_3_); 3.757 ppm(s, 3H, CH_3_); 4.081 ppm(q, 2H, CH_2_); 7.285 ppm(s, 1H, -CH=); 7.362 ppm(s, 1H, -CH=); 8.582 ppm(s, 1H, -CH=). The results for the FT-IR (S1) and NMR (S2) spectra are also provided in the Supplementary Data to confirm compound identity.

### 3.2. Instruments and Procedures

The binary mixture of ([EMIM] [TFA] + TEA) was prepared using a precise electronic balance (±10^−4^ g, Shanghai Precision and Scientific Sky Beautiful Instrument Co. Ltd, Shanghai, China). The viscometer (Zhejiang Jiaojiang Glass Instrument Factory, Taizhou, China) was calibrated in a super constant temperature bath (Shanghai Pingxuan Scientific Instrument Co. LTD, Shanghai, China, ±10^−2^ K) with glycerol, glycol, and water. The viscosity was detected by the Pinkevitch method (according to GB/T 10247-2008), and the viscosity meter was calibrated by glycerol [39,40]. H-NMR spectra were collected on a Bruker AVANCE II 400 M spectrometer using d6-DMSO as the solvent and TMS as the internal standard. FTIR spectra were collected on a Bruker Tencer 27 spectrometer.

The absorption device of NO_2_ is shown in Figure 13. The device mainly includes one homemade NO_2_ generator, an MFC (Tianjin Jisite Instrument Co. Ltd, Tianjin, China, with an accuracy of ±1% F.S.), two sealed flasks, one three-necked flask, one constant temperature reaction bath (DF-101SZ Gongyi Yuhua Instrument Co. Ltd, Gongyi, China, Temperature control accuracy is ±0.1 K), and one temperature sensor.

The three-necked flask was immersed into the constant temperature water bath of a constant temperature magnetic stirrer. NO_2_ was continuously passed into the vessel after its flow rate *v*_NO2_ (*v*_NO2_ = 30 mL·min^−1^) was controlled by MFC to be constant. At the end of the evacuation, quantitative and different proportions of ([EMIM] [TFA] + TEA) solvent were injected into the three-necked flask, and the molar fraction of [EMIM] [TFA] in the mixture was defined as χ_1_ = *n*_[EMIM] [TFA]_/(*n*_[EMIM] [TFA]_ + *n*_TEA_). The mass of absorbent ([EMIM] [TFA] + TEA) absorbing NO_2_ gas and three-necked flask were weighed by an electronic balance at a constant time interval (t = 10 min). Repeat weighing was carried out until the range of mass variation was less than 1%, considered a reaction balance. The NO_2_ load was defined as A_NO_2__= *m*NO_2_/(*m*_[EMIM] [TFA]_ + *m*_TEA_). The unabsorbed residual exhaust was introduced into the sodium hydroxide aqueous solution for recovery. The whole system operated at a constant atmospheric pressure of 0.1 ± 0.005 MPa.

## 4. Conclusions

In summary, the viscosity of binary mixtures of 1-ethyl-3-methylimidazolium trifluoroacetate ([EMIM] [TFA]) + triethanolamine (TEA) was determined, and their application in NO_2_ absorption was studied at 303.15–343.15 K and 1.01 × 10^5^ Pa. At the same time, the ionic liquid [EMIM] [TFA] was characterized by IR and ^1^H NMR spectra. The results showed that the viscosity values of ([EMIM] [TFA] + TEA) are between 9.34–405.92 mPa·s at the measured temperature and the viscosity values decrease with an increase in temperature. At the same temperature, the viscosity values decrease with an increase in [EMIM] [TFA] molar fraction, and the relationship between viscosity and [EMIM] [TFA] molar fraction was in accordance with the sixth-order polynomial equation. The viscosity deviation of ([EMIM] [TFA] + TEA) was negative in the range of measured temperature and mole fraction, and the viscosity deviation of ([EMIM] [TFA] + TEA) binary mixtures decreased with an increase in temperature at the same molar fraction. The fourth-order Redlich–Kister equation well fitted the relationship between viscosity deviation and molar fraction. Moreover, the NO_2_ absorption quantities of ([EMIM] [TFA] + TEA) binary mixtures decrease with an increase in temperature. At the same temperature, the NO_2_ absorption quantities of ([EMIM] [TFA] + TEA) binary mixtures increased with an increase in the ratio of triethanolamine (TEA) in ([EMIM] [TFA] + TEA) binary mixtures, and the relationship between absorption quantities and molar fraction was in accordance with the fourth-order polynomial equation. The experimental results of cyclic absorption showed that the absorption capacity of NO_2_ was almost unchanged through repeated recycling. The results of FT-IR spectra and known literature indicated that the absorption of NO_2_ by ([EMIM] [TFA] + TEA) was a physical and chemical absorption process. Furthermore, the established mathematical model and the optimization calculation by Origin software showed that when the molar fraction of [EMIM] [TFA] was 0.58 at 303.15 K, the corresponding viscosity was 35.51 mPa·s and the absorption amount was 0.7927 g·g^−1^. The ([EMIM] [TFA] + TEA) shows great potential as an alternative absorbent in industrial denitrification due to its sound absorption and low viscosity.

## Figures and Tables

**Figure 1 molecules-26-06953-f001:**
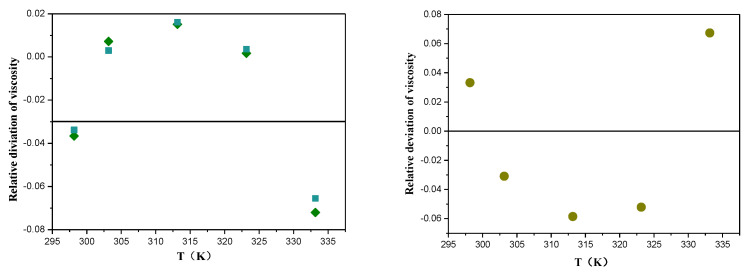
Relative deviation between the viscosity measured in this work and literature values for TEA and [EMIM] [TFA]: (
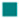
) [33,40]; (
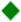
) [34]; (
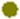
) [35,36].

**Figure 2 molecules-26-06953-f002:**
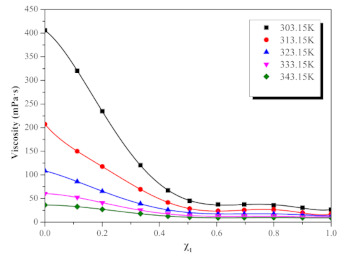
Viscosity η for the ([EMIM] [TFA] + TEA) systems, as a function of mole fraction of ILs, at different temperatures.

**Figure 3 molecules-26-06953-f003:**
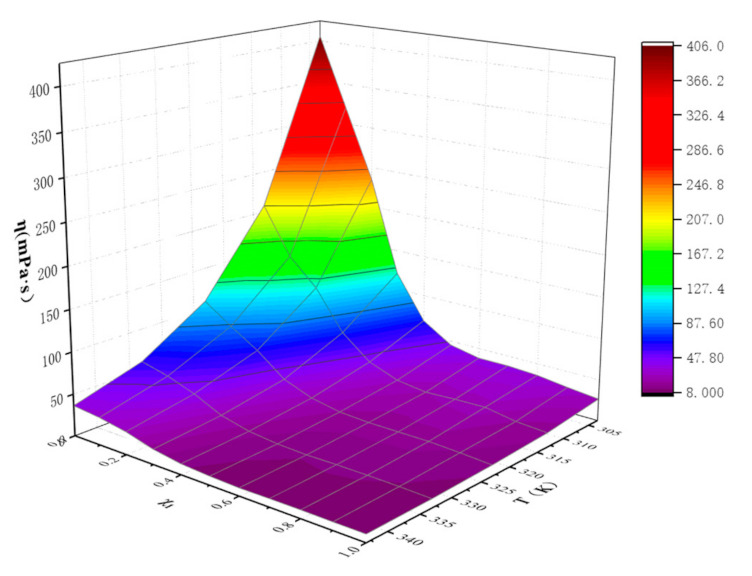
Pseudo-quantitative change of viscosity η with temperature and composition for the ([EMIM] [TFA] + TEA) binary systems.

**Figure 4 molecules-26-06953-f004:**
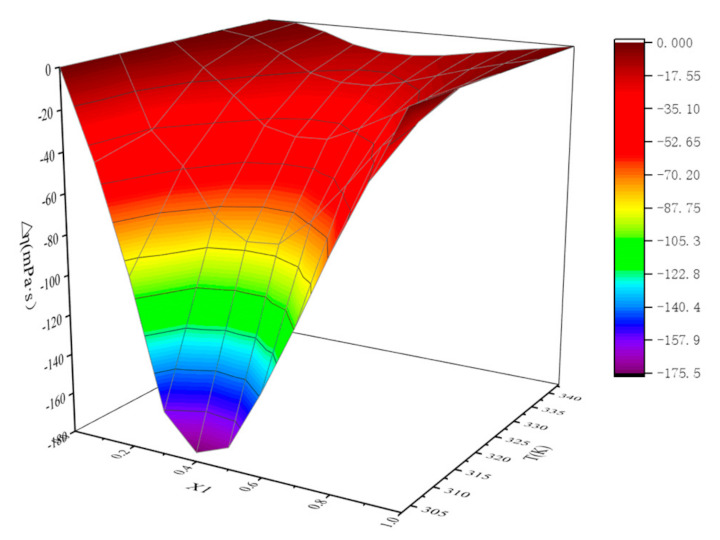
Viscosity deviations of {[EMIM] [TFA] (*χ*_1_) + TEA (1 − *χ*_1_)} binary systems.

**Figure 5 molecules-26-06953-f005:**
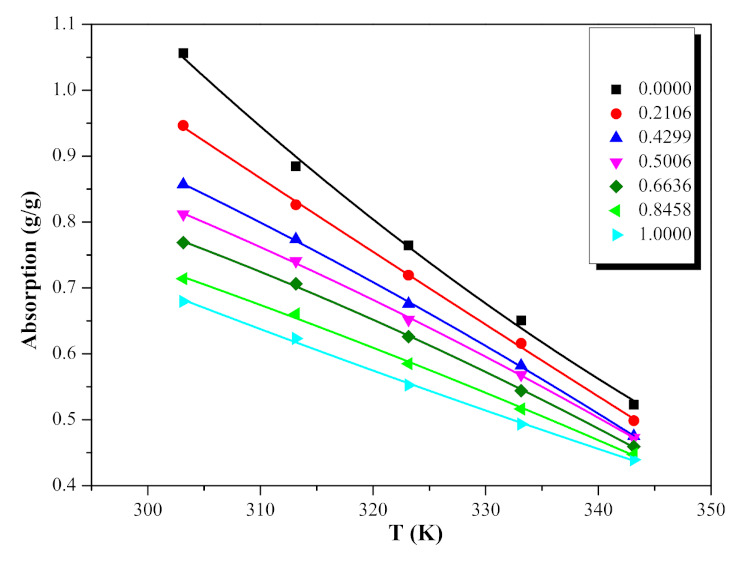
Absorption quantities of NO_2_ for the {[EMIM] [TFA] (χ_1_) + TEA (1 − χ_1_)} binary systems, as a function of temperatures of binary mixture, at different mole fraction of ILs.

**Figure 6 molecules-26-06953-f006:**
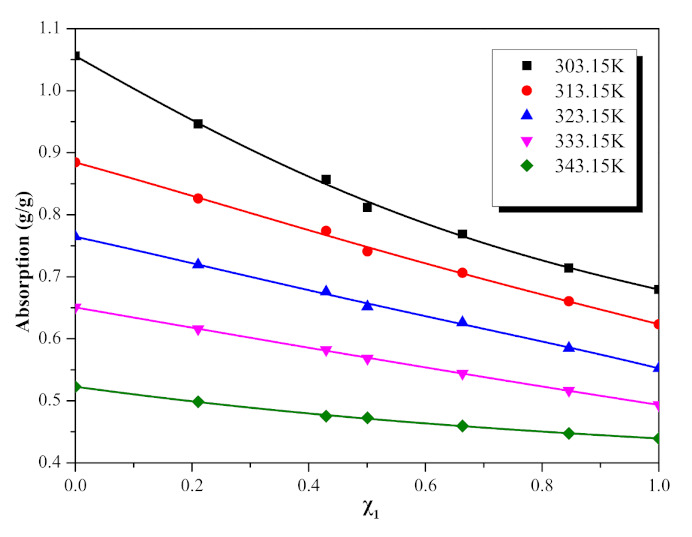
Absorption quantities of NO_2_ for the {[EMIM] [TFA] (χ_1_) + TEA (1 − χ_1_)} binary systems, as a function of mole fraction of ILs, at several temperatures.

**Figure 7 molecules-26-06953-f007:**
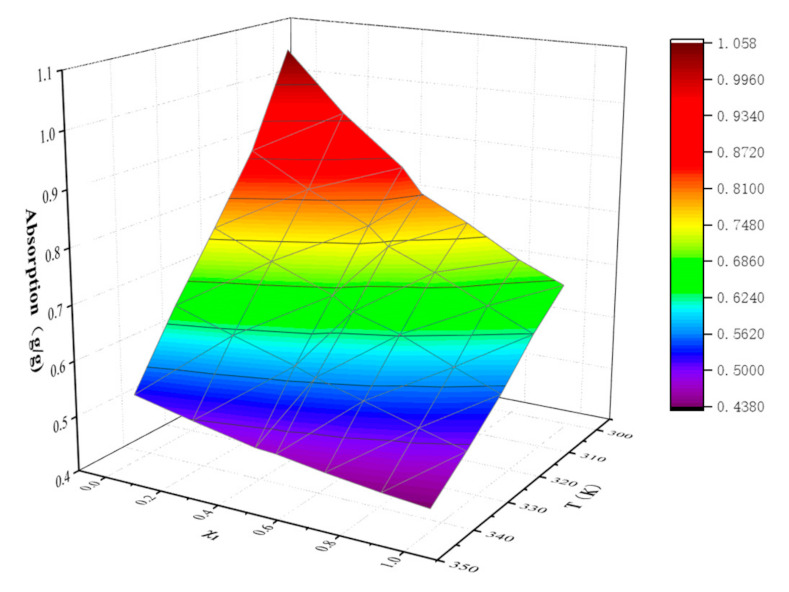
Pseudo-quantitative change of absorption quantities of NO_2_ with temperature and composition for the {[EMIM] [TFA] (*χ*_1_) + TEA (1 − χ_1_)} binary systems.

**Figure 8 molecules-26-06953-f008:**
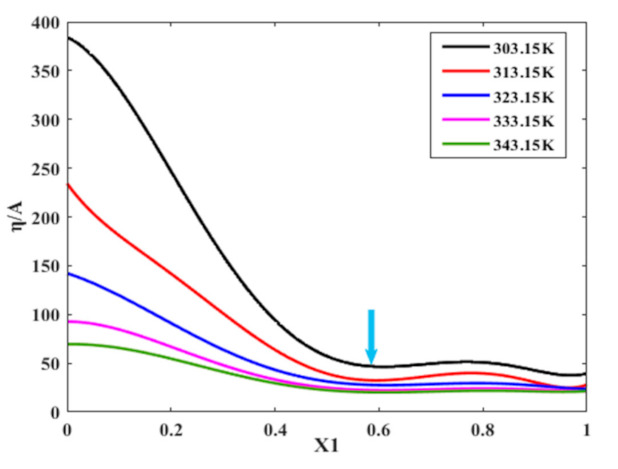
Optimization curves of viscosity/absorption and molar fraction of ([EMIM] [TFA] + TEA).

**Figure 9 molecules-26-06953-f009:**
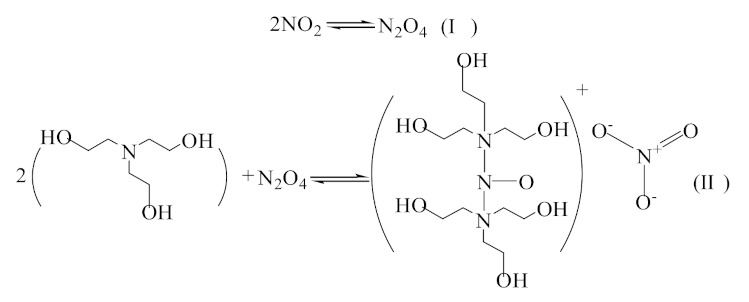
Speculation on reaction mechanism.

**Figure 10 molecules-26-06953-f010:**
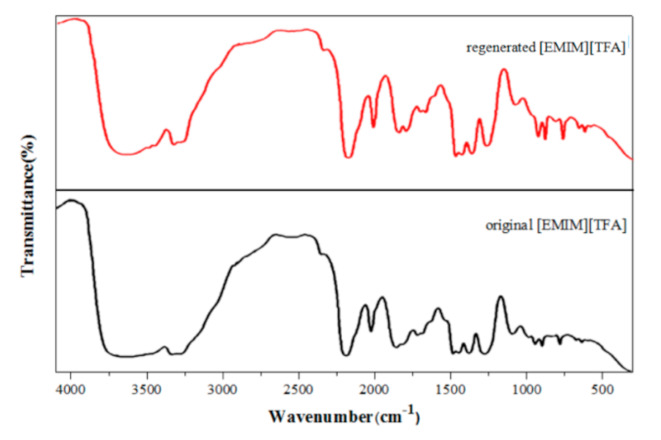
IR spectra of regenerated [EMIM] [TFA] and original [EMIM] [TFA].

**Figure 11 molecules-26-06953-f011:**
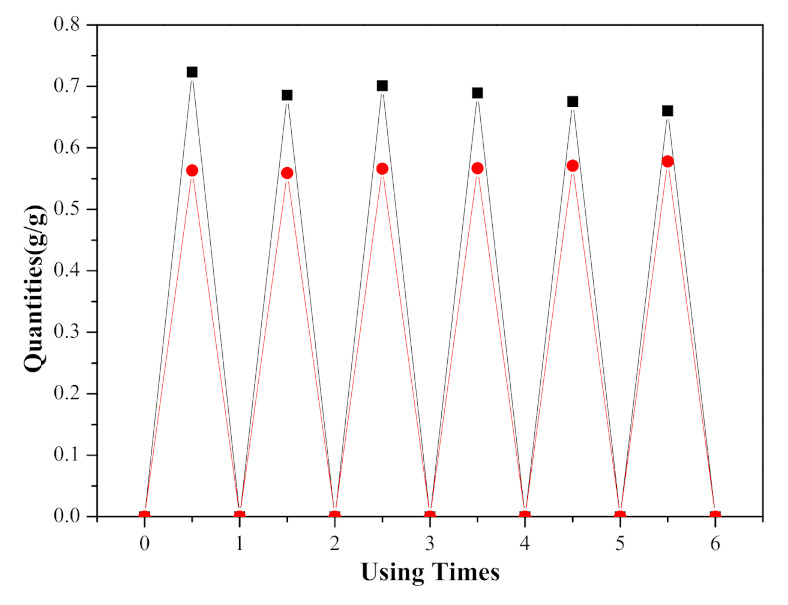
NO_2_ absorption quantities (
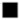
) and desorption quantities (
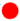
) in ([EMIM] [TFA] + TEA) over six adsorption–desorption cycles.

**Figure 12 molecules-26-06953-f012:**
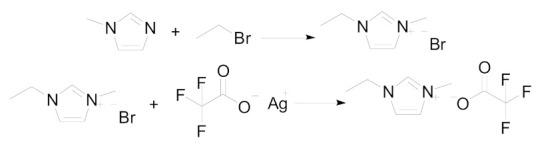
The preparation of [EMIM] [TFA].

**Figure 13 molecules-26-06953-f013:**
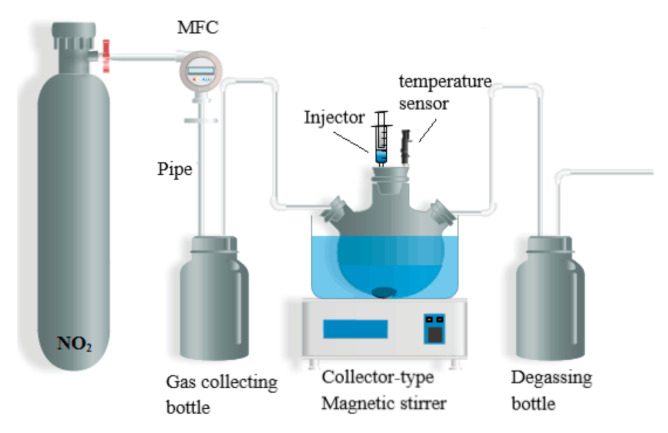
Schematic diagram of experiments.

**Table 1 molecules-26-06953-t001:** Comparison of the viscosity between experiment and literature data ^a^^,b^.

T(K)	TEA Viscosity (mPa·s)			[EMIM] [TFA] Viscosity (mPa·s)	
	Expt	Lit [39,40]	Lit [34]	Expt	Lit [34,35]
298.15	584.98	606.4	604.8	33.10	32
303.15	405.92	403.0	404.7	26.56	27.38
313.15	207.24	204.1	203.9	17.24	18.25
323.15	108.69	108.50	108.3	13.04	13.72
333.15	60.25	64.6	64.2	10.39	11.09

^a^ Standard uncertainties u are as follows: u(T) = ±0.1 K, u(P) = ±0.005 MPa, and the relative standard uncertainties u_r_ in u_r_(η) = 0.27%–4.72%. ^b^ The values in parentheses are the relative uncertainty u_r_(η) × 100% of the measured viscosities.

**Table 2 molecules-26-06953-t002:** Viscosity values of {[EMIM] [TFA](χ_1_) + TEA(1 − χ_1_)} binary systems (mPa·s) ^a,b^.

χ_1_	T(K)				
	303.15	313.15	323.15	333.15	343.15
0	405.92	207.24	108.69	60.25	36.29
0.1128	320.40	149.99	86.10	52.38	32.87
0.2006	234.79	117.78	65.22	40.93	27.13
0.3337	120.61	69.56	39.08	25.55	18.02
0.4299	67.35	41.65	26.19	17.46	12.69
0.5058	44.99	28.61	19.73	13.72	10.47
0.6051	37.59	23.83	17.74	12.32	9.27
0.6965	37.59	25.49	17.60	12.60	9.64
0.7994	35.58	26.28	17.44	12.36	9.74
0.9001	30.63	19.88	15.61	11.64	9.41
1.0000	26.56	17.24	13.04	10.39	9.34

^a^ Standard uncertainties u are as follows: u(T) = ±0.1 K, u(P) = ±0.005 MPa, u(χ_1_) = ±0.0001, and the relative standard uncertainties u_r_(η) = 0.27—4.72%. ^b^ Values in parentheses are the relative uncertainty u_r_(η) × 100% of the measured viscosities.

**Table 3 molecules-26-06953-t003:** The fitting parameters (A_i_) and the determination coefficients (R_2_) of Equation (1) for viscosity of {[EMIM] [TFA] (χ_1_) + TEA (1 − χ_1_)} binary systems at several temperatures ^a^.

T (K)	A_0_	A_1_	A_2_	A_3_	A_4_	A_5_	A_6_	R^2^
303.15	405.84	−460.72	−3557.9	9150.8	−5717.9	−1840.4	2047.1	1.0000
313.15	207.20	−715.74	2963.5	−12302	25291	−23213	7786.5	1.0000
323.15	108.72	−162.30	−475.60	1078.0	157.78	−1399.4	705.82	0.9999
343.15	60.272	−13.505	−674.36	1546.2	−1021.8	−125.96	239.58	0.9999
353.15	36.295	−3.7401	−271.20	206.93	807.81	−1308.2	541.51	0.9999

^a^ Standard uncertainties u are as follows: u(T) = ±0.1 K, u(P) = ±0.005 MPa.

**Table 4 molecules-26-06953-t004:** Viscosity deviations of ([EMIM] [TFA] + TEA) (mPa·s) ^a^.

χ_1_	303.15 K	313.15 K	323.15 K	333.15 K	343.15 K
0.0000	0.00	0.00	0.00	0.00	0.00
0.1128	−42.73	−35.82	−11.80	−2.25	−0.38
0.2006	−95.03	−51.35	−24.28	−9.32	−3.75
0.3337	−158.72	−74.28	−37.69	−18.06	−9.28
0.4299	−175.48	−83.91	−41.38	−21.36	−12.01
0.5058	−169.05	−82.53	−40.58	−21.31	−12.19
0.6051	−138.78	−68.44	−33.07	−17.76	−10.71
0.6965	−104.11	−49.42	−24.47	−12.92	−7.88
0.7994	−67.08	−29.07	−14.79	−8.03	−5.01
0.9001	−33.83	−16.34	−6.99	−3.73	−2.62
1.0000	0.00	0.00	0.00	0.00	0.00

^a^ Standard uncertainties u are as follows: u(T) = ±0.1 K, u(P) = ±0.005 MPa, u(*χ*_1_) = ±0.0001.

**Table 5 molecules-26-06953-t005:** The fitting parameters (*B_P_*) and the standard deviation (*σ*) of Equations (3) and (4) for viscosity deviations of {[EMIM] [TFA] (χ_1_) + TEA (1 − χ_1_)} binary systems at several temperatures ^a^.

T (K)	B_0_	B_1_	B_2_	B_3_	B_4_	σ
303.15	−677.1505	−346.9174	532.6040	536.1765	−127.0621	0.7056
313.15	−331.3255	−125.2933	399.0103	17.7388	−486.4350	0.2340
323.15	−161.1407	−80.3842	129.4112	89.9606	−44.0274	0.2428
333.15	−84.6904	−32.6001	94.7237	74.1388	−14.9678	0.1724
343.15	−49.0278	−7.6149	74.1177	39.5426	−33.6108	0.0761

^a^ Standard uncertainties u are as follows: u(T) = ±0.1 K, u(P) = ±0.005 MPa.

**Table 6 molecules-26-06953-t006:** The absorption quantities of NO_2_ by {[EMIM] [TFA] (χ_1_) + TEA (1 − χ_1_)} binary systems at 303.15 K to 343.15 K (g _NO2_/g_([EMIM] [TFA] + TEA)_) ^a^.

χ_1_	T (K)				
	303.15	313.15	323.15	333.15	343.15
0.0000	1.0562	0.8846	0.7645	0.6506	0.5227
0.2106	0.9465	0.8261	0.7192	0.6159	0.4985
0.4299	0.8570	0.7739	0.6757	0.5820	0.4753
0.5006	0.8119	0.7409	0.6518	0.5683	0.4725
0.6636	0.7687	0.7062	0.6261	0.5440	0.4594
0.8458	0.7139	0.6605	0.5850	0.5163	0.4473
1.0000	0.6795	0.6233	0.5525	0.4933	0.4394

^a^ Standard uncertainties u are as follows: u(T) = ±0.1 K, u(P) = ±0.005 MPa, u(χ^1^) = ±0.0001, u(A_NO2_) = ±0.0001 g.

**Table 7 molecules-26-06953-t007:** The fitting parameters (C_i_) and the determination coefficients (R_2_) of Equation (5) for absorption quantities of {[EMIM] [TFA] (χ_1_) + TEA (1 − χ_1_)} binary systems at different mole fractions ^a^.

χ_1_	C_0_	C_1_ (10^−2^)	C_2_ (10^−5^)	R^2^
0.0000	11.9481	−5.622	6.6857	0.9949
0.2106	5.0106	−1.549	0.6857	0.9987
0.4299	0.5809	1.016	−3.05	0.9996
0.5006	0.1247	1.18	−3.1429	0.9992
0.6636	−0.2954	1.352	−3.3	0.9991
0.8458	0.9571	0.449	1.7429	0.9971
1.0000	3.7355	−1.358	1.1571	0.9971

^a^ Standard uncertainties u are as follows: u(T) = ±0.1 K, u(P) = ±0.005 MPa, u(χ^1^) = ±0.0001.

**Table 8 molecules-26-06953-t008:** The fitting parameters (D_i_) and the determination coefficients (R_2_) of Equation (6) for absorption quantities of {[EMIM] [TFA] (χ_1_) + TEA (1 − χ_1_)} binary systems at several temperatures ^a^.

T(K)	D_0_	D_1_	D_2_	D_3_	D_4_	R^2^
303.15	1.0560	−0.5343	0.0549	0.2016	−0.0987	0.9953
313.15	0.8844	−0.2580	−0.0812	0.1287	−0.0504	0.9941
323.15	0.7645	−0.2040	−0.0712	0.137	−0.0740	0.9955
333.15	0.6506	−0.1629	−0.0040	0.0139	−0.0043	0.9995
343.15	0.5228	−0.1292	0.0577	−0.0096	−0.0021	0.9969

^a^ Standard uncertainties u are as follows: u(T) = ±0.1 K, u(P) = ±0.005 MPa.

**Table 9 molecules-26-06953-t009:** Sample purity description.

Chemical Names	CAS	Level	Purity ^c^	Final Water Mass Fraction	Source
TEA	102-71-6	Analytical reagent	≥0.99	0.0001 ^a^	Tianjin Hedong District Hongyan reagent plant
[EMIM] [TFA]	174899-65-1	-	≥0.99	0.00167 ^b^	Homemade
1-Methylimidazole	616-47-7	Analytical reagent	≥0.98		Shijiazhuang Sydano Fine Chemical Co. Ltd
Silver trifluoroacetate	2966-50-9	Analytical reagent	≥0.99		Tianjin Hines Biochemical Technology Co. Ltd
Bromoethane	74-96-4	Analytical reagent	≥0.99		Tianjin Damao chemical reagent plant
NO_2_	10102-44-0	Analytical reagent	>0.9999		Qingdao Antaike Gas Co. Ltd
Glycerol	56-81-5	Analytical reagent	≥0.99	0.0005 ^b^	Tianjin Yongda Chemical Reagent Co. Ltd
Water	7732-18-5	-			UPH-1V-10T, Chengdu ultra pure water technology Co. Ltd

^a^ Water content of TEA is stated by the merchant. ^b^ water contents of [EMIM] [TFA] and glycerol were measured by Karl Fischer titration. ^c^ Purity of the homemade [EMIM] [TFA] was determined by NMR spectroscopy, and the purity of other chemicals are stated by the merchants.

## Data Availability

Not applicable.

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
