# Peer review of "Experimental Research on NO2 Viscosity and Absorption for (1-Ethyl-3-methylimidazolium Trifluoroacetate + Triethanolamine) Binary Mixtures"

_molecules, 2021, doi:10.3390/molecules26226953_

Round 1
Reviewer 1 Report
On-Page 12: Authors showed the reaction scheme and IR data but don't find any experimental evidence. Authors should show the evidence.
Reviewer 2 Report
In this manuscript authors determined the effects of viscosity and absorption capacity of ionic liquid and triethanolamine binary mixture for NO2 at different temperature range and at different molar fractions. The authors concluded that temperature had no obvious effects on viscosity and absorption capacity while best viscosity and absorption capacity were observed at 0.58 molar fraction of [EMIM] [TFA]. The manuscript is interesting and can be alternative approach as absorbent in the industrial denitrification due to its good absorption and low viscosity. Some comments need to be address before its consideration in molecules
- Rationale for the use of binary mixture of ionic liquid and TFA need to be more emphasize in introduction section.
- Why only one ionic liquid is used in this experiments. Authors need to do the experiments at least three differents type of ILs and optimize with TFA, to choose best one for further experiments. If EMIM is used only then its advantage over other imidazolium based ILs.
- The cost and method complexity based compersion of the the proposed mixture compared to other adsorbents used for NO2 need to present here
Reviewer 3 Report
Review of molecules-1409806-peer-review-v1
Minor comments:
- Typos and grammatical errors are observed, please check and correct.
Major comments:
Introduction:
- Please bring the complete names at the first time of appearance (SCR, ….).
- At the end of the introduction section, some discussions about the results are brought that is not common in writing papers.
Experimental:
- Please introduce the type of the H-NMR and FTIR instruments.
- How the temperature of the water bath was controlled?
- How the setup is connected to infrared spectrometer?
- How the temperature of the solution inside the flask is measured?
- What is the pressure inside the flask? How is the pressure measured?
- Since the necked flask is evacuated first and then filled with the solvent its pressure does not remain constant. Especially after the entrance of the gas the pressure of the flask changes. This important factor is missing. The pressure is not mentioned.
Results and discussions:
- 1- What is the mole fraction of [EMIM] [TFA] and TEA in Tables 4 and 9?
- On page 8 line 229 it is stated “at specific temperature and pressure is shown in Figures 8, 9”. No sign of the amount of pressure is observed in these figures.
- The very important factor that is missing is the pressure inside the flask (the absorption reactor). In the later discussions the authors state that the pressure was 1.01 e+5 pa. The flask is closed and its pressure is not measured. How the authors make such a claim?
- The correlative model used for the absorption amount (the solubility) is very primitive. Please justify the reason behind using this correlation.
Round 2
Reviewer 3 Report
The authors have performed the appropriate changes.
I recommend publication of the manuscript.